# How about an Educational Framework for Nursing Staff in Long-Term Care Facilities to Improve the Care of Behavioral and Psychological Symptoms of Dementia?

**DOI:** 10.3390/ijerph191710493

**Published:** 2022-08-23

**Authors:** Dayeong Kim, Young-Rim Choi, Ye-Na Lee, Won-Hee Park, Sung-Ok Chang

**Affiliations:** 1College of Nursing, Korea University, Seoul 02841, Korea; 2BK21 FOUR R&E Center for Learning Health Systems, Korea University, Seoul 02841, Korea; 3Department of Nursing, The University of Suwon, Hwaseong 18323, Korea

**Keywords:** behavioral and psychological symptoms of dementia, long-term care facility, nursing staff, education, competence

## Abstract

Behavioral and psychological symptoms of dementia (BPSD) are common in residents of long-term care facilities (LTCFs). In LTCFs, nursing staff, including nurses and care workers, play a crucial role in managing BPSD as those most in contact with the residents. However, it is ambiguous where their focus should be for effective BPSD care. Thus, this paper aims to reveal BPSD care competencies for nursing staff in LTCFs and to outline an initial frame of education. A multiphase mixed-methods approach, which was conducted through topic modeling, qualitative interviews, and a Delphi survey, was used. From the results, a preliminary educational framework for nursing staff with categories of BPSD care competence was outlined with the four categories of BPSD care competence: using knowledge for assessment and monitoring the status of residents, individualizing approaches on how to understand residents and address BPSD, building relationships for shared decision-making, and securing a safe environment for residents and staff in LTCFs. This preliminary framework illuminates specific domains that need to be developed for competent BPSD care in LTCFs that are centered on nursing staff who directly assess and monitor the changing and deteriorating state of residents in LTCFs.

## 1. Introduction

The behavioral and psychological symptoms of dementia (BPSD) are a collection of non-cognitive symptoms that include agitation, aggression, apathy, delusion, anxiety, and motor disturbance [1,2]. Unsurprisingly, BPSD care is a major challenge in long-term care facilities (LTCFs), where more than 80% of the residents exhibit more than one symptom [3,4]. With global aging leading to an increasing dementia population, it can be expected that LTCFs and their caregivers need to meet the growing needs of residents who need proper BPSD care [5].

In LTCFs, the nursing staff, including nurses and care workers, undertake a crucial role in managing BPSD as those most in contact with the facility’s residents. Their provision of appropriate BPSD care is imperative to sustaining safe and comfortable lives for LTCF residents [1,6]. Even with the numerous guidelines or strategies available, effective BPSD care cannot be provided if the competence to practice them is insufficient. Because of the difficulties and complexities that the BPSD entail, the importance of nursing staff who can provide competent BPSD care has become paramount [7,8]. However, BPSD care competence has been only treated as a small part of dementia management, which has created new challenges to identifying the nursing staff competencies necessary for delivering proper BPSD care. Clearly identifying competencies according to the different contexts of the client and the practice setting is of particular importance in the nursing discipline as specified competencies can enhance the ability to address nursing phenomena appropriately [9].

Many previous studies have investigated what dementia care competence is. Dementia knowledge and understanding, relationships with residents, person-centered care, and an ethical attitude have all been identified as components of general dementia nursing competence [7,10]. Additionally, in acute care settings, respecting patient individuality and possessing self-facilitation skills have been identified as competence components [11,12]. In LTCFs, promoting an environment that supports independence in daily living and recognizing, preventing, and managing BPSD have been reported as required competencies for dementia care [13]. As BPSD care is found to be a distinct attribute of dementia care in LTCFs, it was also reported that nursing staff’s need for education regarding BPSD care competence is high [8]. However, as we have seen, dominant attention has been placed on dementia care, resulting in BPSD care being hidden in its shadow.

Among previous studies, some proposed education programs have targeted BPSD care. Resnick et al. [14] developed a systematic participatory program that aimed to increase the use of nonpharmacological strategies for BPSD in LTCFs and consisted of multiple strategies, including repeated meetings, education, mentoring, and regular evaluation. The research team provided training for person-centered BPSD care based on the Describe, Investigate, Create, and Evaluate (DICE) method, which is a structured algorithmic approach to the assessment and management of BPSD developed by Kales et al. [1]. Using this approach, Kales, Kern, Kim, and Blazek [15] conducted a one-day training for formal and informal caregivers that aimed to improve BPSD management by dealing with underlying causes through pharmacological and nonpharmacological strategies. These education programs provided instruction to various individuals who engage in BPSD care, such as nurses, care workers, activity staff members, social workers, and family. However, they have not consistently reported positive effects for the education in LTCFs [15,16]. Although their detailed BPSD training system seems promising, the question of where nursing staff should place their focus for effective BPSD care remains unsolved. Those previous studies were not centered on nursing staff and mainly concentrated on delivering and familiarizing their education system to a wide range of caregivers.

Little attention being given to nursing staff and their BPSD care competence has resulted in the insufficient and inconsistent content of relevant education, undermining the quality of BPSD care [17,18]. Continuing BPSD education that has a clear basis on competencies and specifically targets nursing staff would generate consistent positive outcomes for practice and residents in LTCFs [19,20]. Therefore, it is imperative to establish an educational direction suitable for BPSD care in LTCFs that aims to improve the nursing staff’s competence in providing direct care.

Considering the insufficient discussions on BPSD care competence and LTCF nursing staff found in previous studies, appropriate education should have an adequate basis that involves not only existing knowledge but actual practical needs as well. In other words, a systematic study covering theoretical and practical evidence is needed in order to outline a frame of education that will be practically applicable to and highly feasible for LTCFs. This paper thus attempts to reveal nursing staff’s specific competencies required for BPSD care in LTCFs and to sketch out a preliminary educational frame through systematic research procedures.

## 2. Materials and Methods

This study employed a multiphase mixed-methods approach that enables the extensive exploration of a complex subject [21]. A multifaceted investigation can be useful for integrating comprehensive inquiries whose focus moves from general knowledge within the literature to the specific practical knowledge of the nursing staff and then suggests an overview of a preliminary educational framework specific to LTCF nursing staff. 

The research was conducted through three phases. First, the general competence of BPSD care in LTCFs was explored in the existing literature by using topic modeling to identify the key points of BPSD care competence in LTCFs. The practical needs of LTCF education for the BPSD care competence of nursing staff were then identified through qualitative interviews in the second phase. In the last phase, the essential educational components specific to nursing staff were clarified through a Delphi survey. Because of the COVID-19 pandemic situation, we minimized the potential risk of infection by conducting our investigation remotely; interviews were conducted by telephone and surveys were distributed and retrieved by mail.

Each research phase was connected sequentially to increase the logical connection and integration of the findings. Phase 1 was connected to the questions for the interviews of Phase 2, resulting in an integration of the theoretical and practical competence of BPSD care in LTCFs and suggesting items for a preliminary frame of education. These items were validated in Phase 3, which guided an outline of education for improving BPSD competence targeted to LTCF nursing staff. All findings of each phase supported the preliminary educational framework as a final outcome. The overview of the research process is presented in Figure 1.

### 2.1. Phase 1: Exploring Theoretical Evidence

#### 2.1.1. Data Collection

To identify the theoretical evidence of general BPSD care competence in LTCFs, the relevant literature was found in multiple online databases. Search terms were formulated with the related keywords of BPSD, LTCF, and competence. The search was carried out within the title or abstract of the documents, and those written in English were included. After removing duplicates and identifying the topics that populate the available theoretical evidence, all 89 articles were included for topic modeling (Figure 2).

#### 2.1.2. Data Analysis

As BPSD care covers an extensive range of disciplines, including medicine, psychology, and nursing, the traditional method of reviewing the relevant literature would require a great deal of time and labor. Because a machine learning-based literature review can be an efficient and appropriate alternative method [22,23], we used topic modeling. Topic modeling is a technique that can infer latent topics from a large set of documents using computer software that prevents any human biases [23,24], and the topics it extracts indicate the core insights and values of the comprehensive knowledge body [25]. We used the NetMiner 4 program (version 4.4.2.c, Cyram Inc., Seoul, Korea) [26] and followed its guidelines, which consist of five steps: collecting unstructured text data, data pre-processing, analyzing topics, network connection and visualization, and interpreting the results.

From the collected 89 documents, data were pre-processed to extract meaningful words. A node filtering process was set to a word length longer than two letters and a term frequency-inverse document frequency (TF-IDF) equal to or higher than 0.5 [26]. The Latent Dirichlet Allocation (LDA) model was then used to analyze the data and expose hidden topics by categorizing documents and clustering them as topic groups [27]. After the exploratory review of the abstracts of the categorized documents, the first author labeled each topic with a name. Finally, to ensure intra-topic semantic validity, we discussed the coherence of the topic names and the substantive content of the allocated documents by using the following two questions: “Do the topics describe BPSD care competence in LTCFs?” and “Which aspects of the categorized documents describe the issue most comprehensively?” [28].

### 2.2. Phase 2: Identifying Practical Needs

#### 2.2.1. Participants

Based on the theoretical evidence of the general BPSD care competence in LTCFs, specific educational needs for nursing staff were explored through qualitative interviews. Expert in-depth interviews are a method for acquiring deep and extensive insight from experts who can provide fruitful information from accumulated experiences [29]. For this second phase, five experts were purposively recruited. To recruit participants, we contacted experts who had more than ten years of BPSD care experience in LTCFs and explained to them the purpose and procedure of the study. All the contacted experts agreed to participate in the interviews. All participants had more than ten years of experience in BPSD care and represented various occupations, which enabled them to together provide a comprehensive perspective. Three were directors of LTCFs, one was a doctor, and the other two were nurses. A professor of gerontological nursing who had extensive professional careers as both a nurse and director of an LTCF was also included. Lastly, a head nurse in charge of an LTCF unit was recruited for the interviews. The general characteristics of the experts are provided in Table 1.

#### 2.2.2. Data Collection

The interview questions were constructed based on the topics derived in the previous phase. The two questions were: “What education is needed to improve BPSD care competencies of nursing staff in LTCF?” and “What should be taught for nursing staff regarding setting resident-centered goals (Topic 1) in BPSD care?” The interviews were conducted via telephone until they reached saturation for generating abundant empirical knowledge [30]. Each interview took 60 to 90 min and was audio recorded.

#### 2.2.3. Data Analysis

A directed content analysis was used to elicit content essential for BPSD care education. The interview scripts were repeatedly read and coded in their entirety by the first to fourth authors (D.K., Y.-R.C., Y-N.L., and W.-H.P.) according to coding rules using the label and keywords of the topic [31,32]. After coding, the transcribed text was grouped to form categories and abstracted to a higher logical level for organizing essential meanings [32]. Any content that deviated from the codes was excluded. After all of the authors reviewed the coding process together, the preliminary components for the intended frame of education were confirmed.

#### 2.2.4. Trustworthiness

To ensure the trustworthiness of the analysis, the criteria suggested by Lincoln and Guba [33] were used: credibility, transferability, dependability, and confirmability. To assure credibility, all researchers took time to identify and remove any bias within themselves regarding BPSD and LTCFs before starting the analysis. Then we independently categorized the interview transcriptions and compared the results. To establish transferability, we described the discovered categories and sub-categories in detail to sufficiently reflect the meaning of practical experiences. For dependability, all analyzed processes were recorded and reported accurately. To ensure confirmability, we examined our angle of analysis to be certain of the reliability and reflexivity of our results. Finally, the corresponding author confirmed the final results based on the discussions concerning the relevance between the categories and related excerpts. Moreover, we used the consolidated criteria for reporting qualitative research (COREQ) to report the results in Appendix A [34].

### 2.3. Phase 3: Validating Essential Components

#### 2.3.1. Participants

In the final phase, a Delphi survey was conducted to obtain a reliable consensus of nursing staff on whether the preliminary items of the frame were appropriate to improve the BPSD care competence of LTCF nursing staff and were suitable to apply in practice in LTCFs. The participants of a Delphi survey should have expertise in the subject of the survey, although there is no clear criterion for the appropriate number of participants, with the recommendations varying from 10 to 100 [35,36]. In this study, 14 nursing staff members who had more than three years of BPSD care experience in LTCFs were purposively recruited: nine nurses, including two LTCF directors, and five care workers. They worked at three different LTCFs whose capacities range from 56 to 320 residents. The general characteristics of the participants are listed in Table 1.

#### 2.3.2. Data Collection

The survey questionnaire items comprised the categories and sub-categories resulting from the previous phase. A three-round Delphi survey was conducted by mail from August to November 2020, with intervals of two weeks between rounds. In the first round, participants were asked to score the validity and importance of items with a five-point Likert scale (1 = strongly disagree; 5 = strongly agree). In the second and third rounds, the participants were provided with the statistical results of the former round, including the mean, standard deviation, and interquartile range. If their scoring was considered an outlier, participants could provide reasons or add any comments regarding BPSD care education. Similar repeated comments were added as a new item and scored in the next round.

#### 2.3.3. Data Analysis

In addition to the descriptive statistics that were provided to the participants between the survey rounds, we initiated an analysis to identify consensus on the survey questionnaire items. Reaching consensus was defined with two criteria before starting the survey to strengthen the reliability of the consensus [37]. The criteria were (1) more than 80% of the items with a score of four or above and (2) a content validity ratio (CVR) greater than the critical value. Both criteria are the most common ones for when agreement on a certain item is essential [37,38]. According to calculations based on the exact binomial probabilities, the critical value of CVR was 0.571 using the formula [Ne − (N/2)]/(N/2) (Ne = number of participants scoring four or higher, N = total number of participants) [38,39]. Only items that fulfilled the criteria and supported the validity and importance of BPSD care education were included in the final framework.

### 2.4. Ethical Considerations

This study was approved by the Institutional Review Board of the university (KUIRB-2019-0271-03). Every participant in the in-depth interviews and the Delphi survey was provided a written consent form with detailed explanations of the study and assurance of their anonymity.

## 3. Results

### 3.1. Phase 1: Topic Modeling of the Existing Literature

Through the topic modeling of the 89 articles addressing general BPSD care competence in LTCFs, four topics, each with five top contributed keywords, were extracted (Figure 3).

Topic 1, “setting resident-centered goals”, indicated the importance of the nursing staff’s roles in improving residents’ quality of life and well-being by alleviating BPSD and their related symptoms, such as pain, the keyword which had the strongest connection with the topic. Topic 2, “assessing comprehensively”, presents the ability to assess BPSD through using tools and communicating with other staff or family to deal with disturbing behavior symptoms. The studies of Topic 3 emphasized possessing insight into the behaviors of residents and understanding their needs; hence, the topic was labeled “understanding the residents”. As “burden” was found to be the most related keyword, it was identified that the competence of “understanding residents”, notwithstanding its importance, inflicts distress and difficulties on those providing BPSD care in LTCFs. Last, Topic 4, “providing optimal strategies”, contained an investigation of the effects of various strategies to relieve BPSD, including pharmacological and nonpharmacological interventions. Among various strategies, the keyword “medication” was shown to be particularly important in BPSD care in LTCFs.

### 3.2. Phase 2: In-Depth Interviews with Experts

From the expert interviews, four main categories of the practical needs of LTCF education for BPSD care competence were identified (Table 2).

Category 1. Using knowledge for assessment and monitoring the status of residents

For competent BPSD care in LTCFs, not only a solid knowledge base is required, but applying it in practice is also necessary. Based on that requirement, nursing staff should establish personal experiential standards to assess the status of residents and to monitor the administration of medication appropriately.

Assessing residents and their backgrounds holistically

The experts pointed out the importance of assessing residents properly and of knowing exactly what kinds of BPSD are present through comprehensive and holistic assessments. Using tools—one of the keywords of Topic 2—can be a basic capacity, but especially in LTCF practice, it is more necessary to recognize BPSD based on any changes from the holistic data, including the usual behavioral patterns of the residents.

Monitoring pharmacological interventions

The experts mentioned the importance of monitoring pharmacological interventions, such as administrating antipsychotics. Nursing staff need to have accurate knowledge of BPSD medications and how to properly use them when necessary, rather than being hostile to medication due to a fear of side effects. Moreover, the capability of recognizing side effects by monitoring the status of residents and managing them is an important attribute of a competent nursing staff.

Category 2. Individualizing approaches of how to understand residents and address BPSD

Regarding competent BPSD care in LTCFs, the use of an individualized approach has the intention of truly understanding residents, which includes respecting residents and preserving their dignity. The latter element, dignity, can be easily ignored when dealing with residents with BPSD. However, competent nursing staff have to respect and recognize residents as independent individuals who have unique characteristics and the right to sustain a meaningful life in the LTCF. 

Discerning unique individual patterns

All residents have experienced different lives in various socio-cultural environments. Recognizing this contextual background enables a true understanding of residents and improves the discovery of the BPSD’s underlying causes. For this reason, the experts emphasized the nursing staff’s perspective of recognizing a resident as a unique individual who has a personal pattern of living and thinking. To discern these unique individual patterns, nursing staff need to possess a comprehensive knowledge of residents, including their lives before dementia, daily habits, and preferences. Through these data, the quality of BPSD care can be improved via the application of resident-centered personalized care.

Having a consistent and empathetic attitude

The experts indicated that an intimate and empathetic attitude imbued with respect helps residents feel stable and safe. In addition, it was emphasized that, regardless of the nursing staff’s work stress or personal problems, they should maintain a consistent professional demeanor. This consistency positively affects the quality of care and staff members’ relationships with residents.

Adjusting care to individual conditions

The experts underscored the flexibility of care that takes into account the different unique living patterns and physical and mental states of residents with BPSD. A competent nursing staff should not conduct interventions in a uniform and standardized way but rather should take an individualized approach and provide care by adjusting to a resident’s circumstances and context.

Category 3. Building relationships for shared decision-making

Multidisciplinary care, which is a characteristic of dementia care in LTCFs, also plays an important role in BPSD care. However, what is more distinct in competent BPSD care is that it requires a team approach for shared communication to make decisions for best practices.

Sharing views with other staff

The experts stated that sharing views about residents with colleagues helps in obtaining broader insights into BPSD. A deep, comprehensive understanding of BPSD is possible when the perspectives of multidisciplinary staff, such as physical therapists and social workers, are integrated.

Deciding on priorities based on collaborative discussions

According to the experts, discussions with multidisciplinary staff on the causes of and strategies for BPSD are necessary to assure consistent care. Based on such discussions, nursing staff can decide on a top priority and focus on it. In this way, all staff provide care that attends to a resident’s most important problem, maximizing the effect of the BPSD management.

Category 4. Securing a safe environment for residents and staff in LTCFs

As some BPSD may cause harm to residents with BPSD, other LTCF residents, or any staff working in an LTCF, safety should be a central concern of the nursing staff. 

Preventing residents from potential harm caused by BPSD

The experts stated that nursing staff require the competence of securing residents’ physical health by preventing potential harm. Any possible physical injuries directly caused by BPSD, such as falls, fractures, or contusions, should indeed be prevented, but indirect harm, such as malnutrition due to wandering during mealtimes, should also be treated. In addition, it is necessary to detect and remove maintain any dangerous environmental factors that residents with BPSD could come into contact with. 

Considering the safety of all LTCF members

Regarding BPSD care, safety matters to all members of an LTCF, including residents without BPSD and staff. Because BPSD may cause harm to any residents nearby or caregivers, the goals of BPSD care practice should include maintaining everyone’s safety. Furthermore, the experts indicated that safety is the standard for allowing the residents to continue their daily lives within the LTCF. In other words, symptoms can be accepted without intervention if they do not threaten the safety of the resident who exhibits the symptoms, other residents nearby, or staff; otherwise, appropriate interventions should be provided.

### 3.3. Phase 3: Delphi Survey of Nursing Staff Members

The statistical results of each round of the Delphi survey are presented in Table 3.

Although all items met the CVR (0.571), some failed to reach an 80% level of agreement in the first round but reached that level of consensus in the next round. Participants provided many comments about the reasons for their ratings, as well as opinions on additional items. After considering their opinions, one item—“Keeping an up-to-date knowledge of BPSD care”—was added in the second round. In addition, many comments pointed out that education should be focused on nursing practice rather than on delivering information, saying for example, “Education based on cases and examples will be more effective than simple lectures”. After a consensus was reached on all items in the final round, the overview of a preliminary framework of education to improve BPSD care competence in LTCF emerged (Figure 4).

The outlined framework, which is specific to nursing staff and illustrates the core of BPSD care competence in LTCFs, achieved consensus throughout a sequential process of literature review, qualitative interviews, and a Delphi survey. It consists of four main categories of BPSD care competence in LTCFs.

The first category suggests that education is needed to expand the nursing staff’s knowledge base for properly assessing the changing responses of residents and for monitoring their pharmacological management. It focuses on assessing and monitoring the residents and interventions, not on providing treatments for BPSD. The second category underlines the attitudes and approaches for reaching a deep understanding of residents and for dealing with their BPSD. Education for nursing staff should include identifying the patterns that characterize each resident with BPSD and providing consistent care in an empathetic, respectful manner. The third category presents the competence of nursing staff to share information on deciding on common prioritization with the multidisciplinary staff members who engage with residents with BPSD. Nursing staff should establish cooperative relationships and possess effective communication skills to impart information on residents’ complaints and changes that have the potential to cause BPSD, as well as to share best practices. Finally, the last category suggests that nursing staff require the competence to secure the safety of residents with BPSD, other LTCF residents, and staff who provide BPSD care.

## 4. Discussion

This paper has attempted to create a preliminary educational framework which illuminates the core competencies of BPSD care that nursing staff in LTCFs should possess. It is a tentative guideline for education specific to nursing staff in LTCFs. Having nursing staff members as the main subject of the education, our results revealed the domains that need to be improved for competent BPSD care in LTCFs. 

This framework’s specificity is a major difference from previous educational programs in which a nursing staff followed structured steps of assessment and management as a member of a comprehensive caregiver group [1,14,15,16]. The previous studies suggested an integrative approach at the organizational level by presenting overall guidelines for BPSD management. However, as our study focused on nursing staff, investigating their competence in BPSD care, which could improve the quality of direct care in LTCFs.

From the comments collected from the Delphi survey, it was found to be important for nursing staff to make efforts to refine their competencies with more accurate and up-to-date knowledge of BPSD care in LTCFs. This implies the need for nursing staff to incorporate the latest scientific information into their knowledge base as research continues to deliver new information and guidelines for effective BPSD care [6,40]. With the up-to-date knowledge, nursing staff can uncover the personal factors of residents that have the potential to cause BPSD by investigating residents’ prior life experiences, which is important in dementia care in LTCFs [41]. Through this clear and accurate view, nursing staff have the competence of individualized approach for the true understanding of residents with BPSD.

Our results signified that a nursing staff should possess the ability to build a cooperative relationship for collecting comprehensive data, a task they cannot achieve alone. Sharing common priorities and best strategies for BPSD among multidisciplinary staff, including geriatricians, physical therapists, and/or occupational therapists, allows the maintenance of consistent practice, which enables the reduction of changes or confusion in the care process [1]. Therefore, it is suggested that nursing staff-specific education is needed to create an environment where fluent communication for multidisciplinary sharing is possible, such as using the Situation-Background-Assessment-Recommendation (SBAR) technique or a shared mental model [41,42,43].

As BPSD, such as aggression, have a high possibility of harming other residents and staff, it is necessary for nursing staff to be educated about improving perceptions concerning safety in BPSD care. In addition to nonpharmacological interventions with an environmental approach [1,44], a nursing staff-specific education that includes instruction on setting safety-oriented goals for BPSD care is suggested.

It was indicated from additional comments on the Delphi survey that education for nursing staff in LTCFs needs a creative method focusing on a practical approach. Rather than simply conveying theoretical information about BPSD care, it is proposed that BPSD care education should provide more practice-focused instruction, such as case-based or simulation learning [43,45]. When building an education program based on the suggested preliminary framework, more detailed and extended contents could be added. Consulting with multidisciplinary stakeholders in the community, including psychologists or pharmacists, could be considered in making decisions concerning the best practices to be addressed in such a program [16]. Safety issues regarding BPSD care, such as the use of physical or chemical restraints, could also be addressed [9,17]. Moreover, the application of information and communication technologies, such as online-based education that can effectively convey the latest scientific knowledge, could be utilized.

In this study, some limitations should be considered. In the topic modeling process, relevant articles might have been overlooked, but we tried to search as comprehensively as possible by elaborating the search terms and using multiple online databases. Moreover, in the process of categorizing the literature into topics using the LDA model, the extracted keywords may not have accurately reflected a sentence’s context [23]. However, we made concerted efforts to clearly understand the meaning of the topics by reviewing the abstracts repeatedly. Additionally, selection bias could occur during the purposive sampling of the interview and survey participants. To prevent this problem, we were attentive to recruiting participants who could provide various and rich opinions about competent BPSD care and who had successful careers in LTCFs. Lastly, face to face interviews were not conducted to avoid contagion during the COVID-19 pandemic. 

## 5. Conclusions

To sum up, the outlined preliminary framework is concerned with education for improving BPSD care and is centered on the nursing staff members who directly assess and monitor the changing and deteriorating states of LTCF residents. It reflects the core elements of BPSD care in LTCFs that can be utilized as a guide by nursing staff on where and when to be attentive in providing bedside care to residents with BPSD. This preliminary framework also lays the foundation for future work on constructing a systematic and sophisticated BPSD education for LTCF nursing staff. Further research will extend the applicability of the framework, such as the use of online platforms or for coping with special situations, such as the COVID-19 pandemic. Education suitable for social and environmental contexts can be applied more closely to practice, resulting in the improvement of the quality of BPSD care and its positive effects on LTCF residents’ quality of life.

## Figures and Tables

**Figure 1 ijerph-19-10493-f001:**
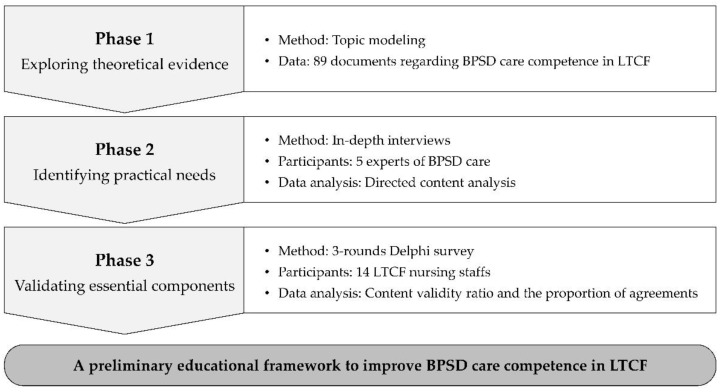
Overview of the research process.

**Figure 2 ijerph-19-10493-f002:**
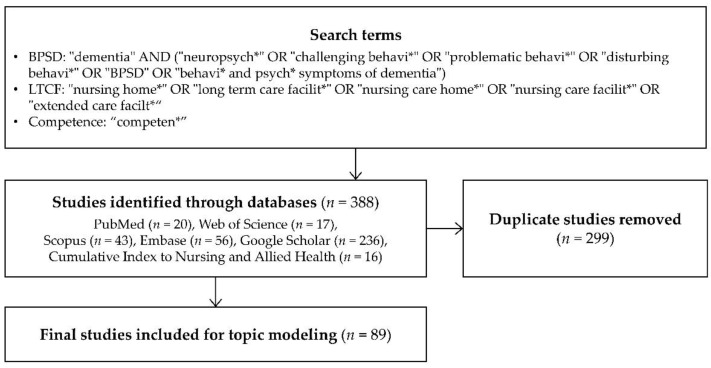
Process of data collection for the topic modeling.

**Figure 3 ijerph-19-10493-f003:**
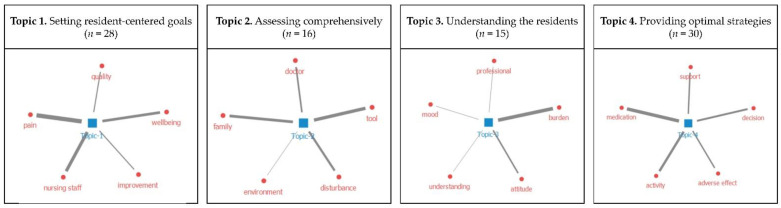
Result of the topic modeling. (Thickness of lines indicates the strength of network between the topic and keywords).

**Figure 4 ijerph-19-10493-f004:**
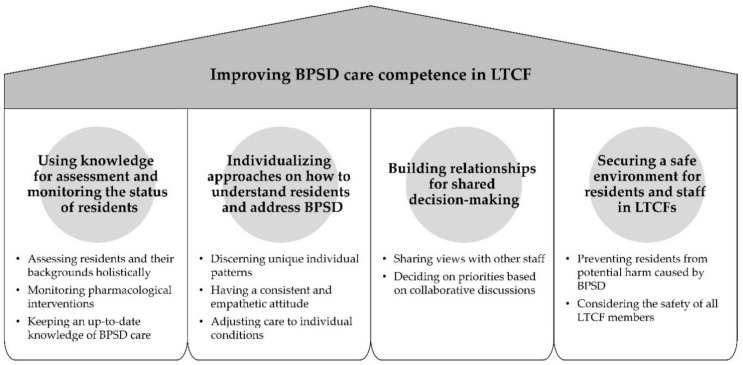
A preliminary educational framework to improve BPSD care competence in LTCF.

**Table 1 ijerph-19-10493-t001:** General characteristics of participants of the in-depth interviews and the Delphi survey.

Variables	In-Depth Interview	Delphi Survey
	(*N* = 5)	(*N* = 14)
Age (M ± SD), years	57.4 ± 8.29	54.5 ± 8.10
Gender (%)		
Male	1 (20)	0 (0)
Female	4 (80)	14 (100)
Profession (%)		
Professor	1 (20)	0 (0)
Director of the long-term care facility	3 (60)	2 (14.3)
Nurse	1 (20)	7 (50)
Care worker	0 (0)	5 (35.7)
Education level (*%*)		
Graduate school	4 (80)	4 (28.6)
University	1 (20)	6 (42.8)
High school	0 (0)	4 (28.6)
Working experience (M ± SD), years	26.32 ± 9.72	18.02 ± 11.68

(M ± SD: Mean ± Standard Deviation).

**Table 2 ijerph-19-10493-t002:** Categories of the practical needs of LTCF education for BPSD care competence with relevant statements from the in-depth interviews.

Category	Sub-Category	Relevant Quotations from the Interviews
I. Using knowledge for assessment and monitoring the status of residents	1. Assessing residents and their backgrounds holistically	*It is important to get information from their families to understand the residents. Based on that, we can figure out the underlying causes of BPSD. It needs efforts to draw out useful information from communication with them.* (Expert 2)
	2. Monitoring pharmacological interventions	*The most important thing when using medications is to check the safety of the residents rather than the symptoms themselves. Although medications are important in managing BPSD, side effects should be carefully considered. It is very important to balance the benefits and side effects of drugs.* (Expert 1) *I use medications as long as I see residents’ pupils stay clear. If their eyes are blurred as if they were drunk, it means the medication is not appropriate and some action is needed. Non-pharmacological interventions are used if the symptoms do not harm other residents.* (Expert 3)
II. Individualizing approaches on how to understand residents and address BPSD	1. Discerning unique individual patterns	*Residents with BPSD do not express their needs in an ordinary way. In that case, we need to read their minds to understand their needs by a combination of data such as their personal characteristics and preferences.* (Expert 3) *Even if they have exactly the same symptoms, each resident has different life experiences. The only way to find the underlying causes of BPSD, nurses need to investigate residents’ life experiences, habits, and values.* (Expert 4)
	2. Having a consistent and empathetic attitude	*The respectful attitude of the nursing staff also affects other staff’s care. It is professional care to maintain a consistent attitude of respect and interest toward residents.* (Expert 4) *Even if they have dementia, they know who truly cares sincerely for them. They give more trust to those sincere nurses and follow their instructions better.* (Expert 5)
	3. Adjusting care to individual conditions	*Nursing staff should consider the condition of the symptoms, and not always provide the same care. On days when residents are having a hard time or bad condition, we need the flexibility to make it an alternate approach even if providing the same program.* (Expert 5)
III. Building relationships for shared decision making	1. Sharing views with other staff	*Nursing staff should check the resident’s diet, sleeping, body movements, and everything in detail by asking other staff and sharing information, and taking a holistic approach to the resident.* (Expert 3) *It is important to listen to other staff’s opinions. It is good to use tea-time to talk about residents’ daily conditions and any information about BPSD.* (Expert 4)
	2. Deciding on priorities based on collaborative discussions	*I think the most important point is finding the symptoms that many practitioners feel are problematic in common. Those various perspectives of interdisciplinary staff give nursing staff a comprehensive insight into the utmost important care residents need to alleviate BPSD.* (Expert 2)
IV. Securing a safe environment for residents and staff in LTCFs	1. Preventing residents from potential harm caused by BPSD	*Nursing staff should not miss the possibility of physical harm in managing symptoms. It is important to check carefully if residents have lunch enough or if they bumped into anything. So much detailed management is needed, such as giving finger food to prevent nutritional deficiencies due to BPSD.* (Expert 2)
	2. Considering the safety of all LTCF members	*Always be aware that BPSD can affect the safety of the entire LTCF members as well as the residents with BPSD. It is also up to the nursing staff to maintain a safe LTCF community where there is no harm to other residents, and our staff can provide care safely.* (Expert 1) *If symptoms do not harm the resident oneself or others, it is enough to express their symptoms freely as long as they are physically safe. They can be left alone without being taken any special care.* (Expert 4)

Expert 1: M.D., Director of LTCF; Experts 2 and 3: RN, Director of LTCF; Expert 4: Professor of gerontological nursing; Expert 5: a head nurse.

**Table 3 ijerph-19-10493-t003:** Result of the Delphi survey.

Category	Sub-Category	1st Round						2nd Round						3rd Round					
		Validity			Importance			Validity			Importance			Validity			Importance		
		M ± SD	%	CVR	M ± SD	%	CVR	M ± SD	%	CVR	M ± SD	%	CVR	M ± SD	%	CVR	M ± SD	%	CVR
I. Using knowledge for assessment and monitoring the status of residents	1. Assessing residents and their backgrounds holistically	4.42 ± 0.93	85.71	0.714	4.78 ± 0.57	92.85	0.857	4.64 ± 0.63	92.85	0.857	4.92 ± 0.46	100	1	4.71 ± 0.75	92.85	0.857	4.92 ± 0.26	100	1
	2. Monitoring pharmacological interventions	4.35 ± 0.84	78.57 ^†^	0.571	4.71 ± 0.61	92.85	0.857	4.42 ± 0.64	85.71	0.714	4.78 ± 0.42	100	1	4.78 ± 0.63	92.85	0.857	4.78 ± 0.26	100	1
	3. Keeping an up-to-date knowledge of BPSD care ^‡^	-	-	-	-	-	-	4.28 ± 0.82	78.57 ^†^	0.571	4.42 ± 0.75	85.71	0.714	4.50 ± 0.75	85.71	0.714	4.57 ± 0.63	92.85	0.857
II. Individualizing approaches on how to understand residents and address BPSD	1. Discerning unique individual patterns	4.35 ± 0.84	78.57 ^†^	0.571	4.64 ± 0.63	92.85	0.857	4.64 ± 0.63	92.85	0.857	4.64 ± 0.49	100	1	4.85 ± 0.75	100	1	4.85 ± 0.53	100	1
	2. Having a consistent and empathetic attitude	4.57 ± 0.64	92.85	0.857	4.57 ± 0.64	92.85	0.857	4.57 ± 0.64	92.85	0.857	4.50 ± 0.65	92.85	0.857	4.85 ± 0.53	100	1	4.78 ± 0.26	100	1
	3. Adjusting care to individual conditions	4.35 ± 0.84	78.57 ^†^	0.571	4.64 ± 0.63	92.85	0.857	4.64 ± 0.63	92.S85	0.857	4.78 ± 0.42	100	1	4.78 ± 0.53	100	1	4.92 ± 0.57	100	1
III. Building relationships for shared decision-making	1. Sharing views with other staff	4.71 ± 0.61	92.85	0.857	4.64 ± 0.63	92.85	0.857	4.85 ± 0.53	92.85	0.857	4.78 ± 0.57	92.85	0.857	4.92 ± 0.74	100	1	4.85 ± 0.46	92.85	0.857
	2. Deciding on priorities based on collaborative discussions	4.64 ± 0.63	92.85	0.857	4.85 ± 0.53	92.85	0.857	4.74 ± 0.61	92.85	0.857	4.85 ± 0.36	100	1	4.78 ± 0.89	92.85	0.857	4.92 ± 0.51	100	1
IV. Securing a safe environment for residents and staff in LTCFs	1. Preventing residents from potential harm caused by BPSD	4.57 ± 0.75	85.71	0.714	4.78 ± 0.57	92.85	0.857	4.78 ± 0.57	92.85	0.857	4.85 ± 0.53	92.85	0.857	4.85 ± 0.64	92.85	0.857	4.85 ± 0.63	92.85	0.857
	2. Considering the safety of all LTCF members	4.14 ± 0.94	78.57 ^†^	0.571	4.42 ± 0.75	85.71	0.714	4.50 ± 0.75	85.71	0.714	4.78 ± 0.42	100	1	4.64 ± 0.63	92.85	0.857	4.92 ± 0.63	100	1

^†^ = Under the 80% of agreement; ^‡^ = Newly added items after participants’ comments. (M ± SD: Mean ± Standard Deviation; CVR: Content Validity Ratio).

## Data Availability

Data that support the findings of the study are available upon reasonable request from the corresponding author.

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
