# Peer review of "How about an Educational Framework for Nursing Staff in Long-Term Care Facilities to Improve the Care of Behavioral and Psychological Symptoms of Dementia?"

_ijerph, 2022, doi:10.3390/ijerph191710493_

Round 1

Reviewer 1 Report

I have read the entire material, this is very well designed and thoughtful research on important aspects of practice in long-term care, the aspects covered are multi-disciplinary and the assessed team should include a psychologist or psychiatrist, the research design and methodology are clear, the results are clear. Please only indicate in the methodology what statistical tests were used and describe the limitations of the study

Reviewer 2 Report

I enjoyed reading your paper. The following are feedback and comments for your consideration:

Originality:  As the authors argued, there are limited studies examining BPSD care competencies for nursing staff in LTCFs. Therefore, this manuscript demonstrates adequate originality that justifies the consideration for publication.

Relationship to Literature: The manuscript generally demonstrated a sufficient understanding of the relevant literature through citing and discussing an appropriate range of literature sources that examined existing study findings and their theoretical underpinnings. This manuscript has been developed based on a solid literature review, and its findings have been discussed in the context of literature as well as provide additional information to the current knowledge base.

Methodology: The methods on which this manuscript’s framework was built were overall appropriate and sufficiently described. It however would be helpful, if the authors could further clarify what inclusion and exclusion criteria were used to select studies for analysis in the first phase.

Results and conclusions:  The results were overall presented clearly and analyzed appropriately. The discussion and conclusion also seemed to adequately tie together the other elements of the manuscript. 

Implications: Implications for the nursing profession were not clearly discussed. It would be helpful if the authors could further discuss the implications for practice, research, and education.

Quality of Communication: The manuscript was very well written. But it could use further proofreading and editing as there were some issues with headings and citations. For example, in line 129, the heading should have been 2.1.2 Data Analysis rather than 3.1.2.

Reviewer 3 Report

Thank you very much for allowing me to read this interesting manuscript. The manuscript is based on a topic of great interest due to the impact in nurse and care education and could contribute to the knowledge about BPSD. I appreciate a well-written paper with a strong background. Nevertheless, there are some comments and recommendations that I would like to make:

·       The paper is very ambitious and the three parts are of great interest, however, condensing them into a single manuscript probably makes them lose potential and lacks a solid methodology.

·       In the introduction, and in general throughout the document, there is a lack of reference to the use of physical restraints and their necessary environmental modifications to favour the control of BPDS in the area of patient safety.

·       In this sense, and in reference to the multidisciplinary team, although it is explained in a general way, references to other professionals such as geriatricians, psychologists or, especially, occupational therapists (because it is the health profession specialised in contexts and environments, in promoting independence in daily life and whose approach is always bases in client-centered) are lacking.

·       In the literature review conducted, after discarding duplicates, did all the articles located meet the inclusion criteria and were they consistent with the topic you were researching? What were the inclusion criteria?

·       What kind of review have you done, exploratory, theoretical? This point should be clarified.

·       89 documents seems like a manageable number of paper for a team of researchers, so why did they choose to use computer software? They were able to avoid human bias, but what are the limitations of such software?

·       How Phase 1 and Phase 2 sampling was conducted.

·       It would be interesting to include the script of the interviews in order to understand the answers more easily. Was any quality index used or was only the trueworthiness analysis applied?

·       Data analysis technique: it is not detailed whether any software or computer support was used for qualitative research to facilitate the triangulation of data and the generation of categories. It is deduced that the interviews were transcribed, but this is not indicated. In all the processes they refer to the plural (together), but it is not known how the work was distributed and how many authors participated at each moment.

·       As indicated above, there is also a gap in the discussion when talking about staff or lack of discussion and references to a very important issue such as restraint reduction, etc.

·       “Because of the 106 COVID-19 pandemic situation, we minimized the potential risk of infection by conducting 107 our investigation remotely; interviews were conducted by telephone and surveys were 108 distributed and retrieved by mail.” Any limitations about this?
